# Effects on Taxiing Conflicts at Intersections by Pilots' Sensitive Speed Adjustment

**Kai Yang** [1,2,*], **Hongyu Yang** [1,2], **Jianwei Zhang** [1,2] **and Rui Kang** [3]

1   Institute of Image & Graphics, College of Computer Science, Sichuan University, Chengdu 610064, China; yanghongyu@scu.edu.cn (H.Y.); zhangjianwei@scu.edu.cn (J.Z.)
2   National Key Laboratory of Fundamental Science on Synthetic Vision, Sichuan University, Chengdu 610064, China
3   School of Air Traffic Management, Civil Aviation Flight University of China, Guanghan 618307, China; kangrui@cafuc.edu.cn
*   Correspondence: yangkai@scu.edu.cn

**Abstract:** The pilot is the main person in charge of taxiing safety while moving on the airport surface. The visual separation and speed adjustment are directly related to safety and efficiency of airport surface operation. According to the actual taxiing procedures and airport control rules in China, this paper proposes a novel microscopic simulation model based on the pilots' visual separation. This model is also built by refining the aircraft taxiing procedures at intersections. The observation range, the separation judgment, pilots' visual distance, rate of proximity and the intention for speed governing are discussed as parameters in the model. The rules for aircraft separation judgment, pilots' autonomous speed governing, and position updates are also set up and discussed. The proposed simulation can accurately simulate the acceleration and deceleration intentions under different motion trends while reproducing the motion process including the following acceleration, following deceleration and delayed deceleration caused by separation changes. The results demonstrate that the number of conflicts can be reduced to 50% based on visual separation adjustment of 50 s when the convergence angle is 30°. The pilot's visual distance is inversely proportional to the fluctuation range of the speed of the rear aircraft, the proximity rate of the front and rear aircraft and the probability of conflict. The simulation results of this model conform to the actual taxiing routes and control rules, which provides technical support for improving the safety level of airport surface operation and presents certain reference value and practicability.

**Keywords:** airport traffic management; taxiing conflict; visual separation; computer simulation; numerical modeling; pilot; human performance model

## 1. Introduction

With an increasing number of flight demand, the operation on the airport runways and taxiways is very busy [1]. The complex structure of the taxiways leads to a convergence trend during taxiing and the reduced horizontal separation, making it prone to unsafe accidents such as taxiing conflicts, congestion and even collisions [2]. It has always been a hot issue for civil aviation traffic safety to study the aircraft surface operation process, detect the taxiing conflicts and quantify and evaluate the risk probability [3].

To reduce the taxiing risk, many researchers have focused on the resource utilization optimization on airport surface. In 2011, Li et al. [4] proposed an improved A* algorithm to solve taxiing routing problem by considering taxiing distance and surface topology. In 2015, Tang et al. [5] studied a novel algorithm for aircraft taxiing optimization based on the idle time windows and adopted the Advanced Surface Movement Guidance Control System (A-SMGCS) to implement the algorithm, where the average aircraft taxiing time was reduced by 19.6%. In 2018, Jia et al. [6] proposed a taxiing optimization model based on aircraft priority and conflict resolution strategy with the objective of minimizing operation

costs. Compared to a fixed taxi path based on a First-Come-First-Served (FCFS) principle, this method can reduce the taxiing distance and time. In 2020, Soltani et al. [7] presented a hybrid taxiing solution to reduce the greenhouse gas emissions and avoid taxiing conflicts. The above methods have built up a strategy to optimize the taxiing of aircraft with the aim of optimizing the key resources allocation. Its effectiveness relies on the accuracy of the prediction of when the aircraft will reach the critical resources. If the airport is not equipped with surveillance equipment, it is impossible to obtain the real-time speed and position of the aircraft and accurately predict when the aircraft will reach the critical resources. If the airport lacks a control and guidance system, the controller needs to use instructions to strictly control the time during which the aircraft arrives at the key resources, resulting in the increased control load and operational risks [8].

Some researchers have explored the taxiing process of aircraft to improve the accuracy of aircraft trajectory prediction. In 2013, Ravizza et al. [9] proposed a model to predict the taxiing time by analyzing the previous taxiing data and taking the taxiing interruptions and taxiway changes into consideration. In 2015, Zhou [10] first calculated K taxiing route sets between each pair of stand and runway, and then based on the nominal velocity profile to predict the taxiing trajectory of each aircraft. In 2018, Son et al. [11] studied a control method to keep the aircraft taxiing along the centerline in the case of oversteering of the nose wheel and poor surface conditions. In 2019, Krawczyk et al. [12] explored the tire-ground contact model to abstract the taxiing process of the aircraft and calculate the critical speed of taxiing on the different taxiways. Overall, the above studies only focused on a single aircraft movement. To date, a few research papers has taken interaction movement into account, despite of the fact that multiple aircraft on the taxiways simultaneously will affect each other and the taxiway configuration and operation rules have a great influence on the taxiing speed and time. Therefore, many complications exist in the application of the above models in actual operation.

Existing research have widely concentrated on the aircraft taxiing model based on the characteristics of the surface structure and the control rules. In 2013, Mori et al. [13] designed an aircraft taxiing cellular automata model based on the Nagel-Schreckenberg (NS) model. It was verified with the airport ground monitoring data of Tokyo International Airport and its average accuracy was about 30 s. In 2014, Zhang et al. [14] designed a multi-agent model consistent with the taxiing rules and the controllers' experience. They also described the state evolution process of Agent by using event-condition-action language and verified the Agent model with Anylogic simulation platform. Cetek et al. [15] adopted fast real-time simulation technology to simulate the traffic flow in the motorized area in two stages and analyzed the formation of congestion points. In 2016, Yang et al. [16] introduced the cellular automata (CA) model into the taxiing and rolling process of the aircraft to quantify the effect of the taxiing speed on the take-off and landing intervals. In 2018, Xin et al. [17] proposed an airport ground traffic simulation method based on agent theory and cellular automata model and demonstrated the characteristics of airport traffic flow. Kang et al. [18,19] analyzed the effect of rapid exit from the runway on improving the efficiency of runway use by illustrating the taxiing process of the aircraft in the area where the runway and the taxiway meet. In 2019, Xue et al. [20] combined cell transmission model (CTM) with taxiway operation rules to establish a taxiway traffic flow unit transfer model and employed the NetLogo platform to simulate the evolution of traffic flow at a large airport in China. Yang et al. [21] modeled the taxiing process of the aircraft under low visibility, which can evaluate the interval changes during real-time CAT IIoperations at the airport. In 2021, Kang et al. [22] modeled the taxiing process of aircraft at an airport with mixed operation mode and short-distance parallel runway, where the airport capacity was quantified using taxi time. The above studies can reflect the macroscopic characteristics of the traffic flow by simulating the microscopic motion of the aircraft while reproducing the smooth, congested, or self-organized state of the ground traffic. However, in actual airport surface operations, the controllers cannot obtain the precise taxiing speed in real time and can only estimate the intervals and risk probability based on observation and experience since most airports

are not equipped with surface guidance and control systems [23]. Therefore, the controllers seldom stipulate the taxiing speed, do not issue the speed regulation command frequently and only remind the pilot to observe the traffic when the pilot does not take any measures under the conditions of the chasing or converging trends [24]. The pilot is the main person in charge of taxiing safety. They are responsible for keeping a visual separation by visualizing the conflicted aircraft and the taxiway structure ahead [25]. In recent years, network based trajectory prediction is proposed [26,27]. Some researchers have studied the enhanced cockpit equipment such as taxi electronic map system [28], head-up display [29] and cockpit taxi control system [30], which provide solutions for pilots to maintain visual observation and keep visual separation under the conditions of crowded traffic and low visibility. Jon B et al. [31] examined pre-takeoff control surface checks in taxiing aircraft. Evidence of pre-takeoff control surface checks were identified in FOQA data for departures at Barcelona-El Prat airport by looking for consecutive full-range motion in rudder angle, aileron angle, and elevator angle for aircraft on the airport surface. NASA designed an eXternal Vision System (XVS) that, with other aircraft systems and subsystems, created an electronic means of forward visibility for the pilot [32]. It can be revealed that the pilot's visual observation, interval judgment and autonomous speed control operation are the crucial steps in the taxiing process.

However, the changes in aircraft speed in the above models are relatively monotonous. Rules for acceleration and deceleration are defined by the minimum taxiing separation or the controller's instructions, which reduces the effect of the pilot's driving intention. To assess the effect of the pilot's intentions and actions on the risk of conflict becomes impossible. In order to explore the details of the taxiing process and the risk evolution trend of the aircraft, the pilot's visual separation procedure should be introduced and the pilot's separation judgment and autonomous speed regulation behavior, the construction of the taxiing speed and the dynamic change rules of the positions should also be considered. To address the research gap, this paper proposes a novel microscopic simulation model based on the pilots' visual separation for airport surface operation. This is achieved by considering the actual taxiing procedures and operational rules [33,34], Innovation are (1) the pilot visual distance is introduced, (2) aircraft proximity rate are defined under the illustration of pilot observation range, separation judgment and speed control intention, (3) the taxi speed and position update rules are also established in this model.

The Visual C++ platform is adopted to implement the model for analyzing the influence of the key parameters on the separation, taxiing speeds and conflict probability. The influence of key parameters such as pilot visual distance on the taxiing efficiency and safety level is investigated by using the simulation data and actual operation data. Additionally, the influences of pilot visual distance and autonomous speed regulation behavior on risk level are discussed under the different visual conditions. It is expected to provide solutions for reproducing the movement situation of the airport surface, predicting the airport operation conflict and evaluating the pilot's observation ability, and operation level.

## 2. Rules for Visual Separation Establishment

Figure 1 shows a taxiing merge process, where aircraft $f_i$ and $f_j$ are heading to the same intersection. As suggested in this figure, the dark gray part indicates the taxiway intersection, the dotted line with the arrow indicates the taxiing path and moving direction of $f_i$ and $f_j$, with the intersection point of taxiing paths as $O_{i,j}$. At time $t$, the speed of the two aircraft is $v_i(t)$ and $v_j(t)$ respectively, with the distance from the intersection point being $L_i(t)$ and $L_j(t)$ and with the distance from the intersection point as $g_i(t)$ and $g_j(t)$. The visual range of pilots is $R_i$ and $R_j$, with the light-colored fan area of $160°$ in front of $f_i$ and $f_j$ as the visual range of pilots. The lateral separation between the two aircraft is $D_{i,j}(t)$ (shown in Figure 1). Both visual ranges cover the dark-colored intersections, suggesting that the pilots of the two aircraft can find the intersection point of front taxiways with convergence trends and find the potential conflicts by left-right observation. Considering that $f_i$ is within the visual range of the pilots of $f_j$, the pilots of $f_j$ can observe the distance

to the position of $f_i$ and decide whether it meets the requirement of safe separation for speed adjustment. However, $f_j$ is within the visual range of the pilots of $f_i$, the pilots of $f_i$ can only regulate the speed or keep taxiing at a constant speed according to the conditions such as the configuration and path of the taxiway.

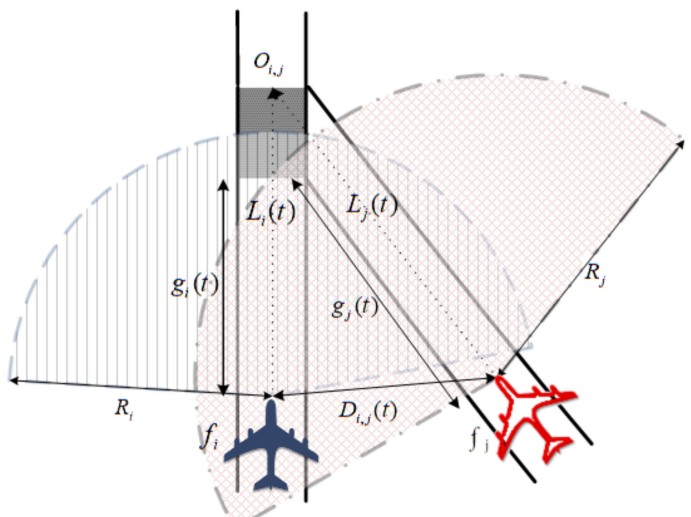

**Figure 1.** Rules for Visual Separation Establishment in Aircraft Taxiing.

Following the above rules, the establishment of visual separation during taxiing should meet the two requirements simultaneously: (1) the pilots can see the intersection and know the tendency of convergence; (2) the pilots of the following aircraft (hereinafter referred to as the following aircraft) can observe the proceeding aircraft (hereinafter referred to as the proceeding aircraft), as well as its taxiing speed and direction.

## 3. Simulation Model Set-Up

### 3.1. Model Constraints

This section provides a concise and precise description of the experimental results, their interpretation and the experimental conclusions.

At time $t$, whether the following conditions are met should be considered according to the position and distance of $f_i$ and $f_j$:

$$D_{i,j}(t) \leq R_j \tag{1}$$

$$g_j(t) \leq R_j \tag{2}$$

If condition (1) is not met, $f_j$ is not within the visual range of $f_i$, it should be gradually accelerated to the comfortable taxiing speed of $v_C$ at an allowable acceleration [13]:

$$v_j(t+1) = \min(v_C, v_j(t) + a_v) \tag{3}$$

If condition (2) is not met, the intersection cannot be observed

$$g_j(t) > \frac{(v_j(t))^2 - (v^T)^2}{a_v} \tag{4}$$

where the speed of $f_j$ is not changed; otherwise, the aircraft gradually accelerates or decelerates to the turning speed limit of $v^T$ following the comfortable acceleration of $a_v$:

$$v_j(t+1) = \max(v^T, v_j(t) \pm a_v) \tag{5}$$

where the value of $v_C$ and $v^T$ is determined by the taxiway width, turning angle and wingspan of $f_j$ [18,29].

If both conditions (1) and (2) can be met, $f_i$ and the intersection can be observed by $f_j$, and pilots of $f_i$ can establish the visual separation as per the position of $f_i$.

### 3.2. Proximity Rate Calculation

The longitudinal separation at the intersection $O_{i,j}$ between $f_j$ and $f_i$ at $t$ is determined as:

$$S_{i,j}(t) = L_j(t) - L_i(t) \tag{6}$$

If $S_{i,j}(t) < 0$, $f_j$ is closer to $O_{i,j}$. According to the principle of priority order, $f_j$ shall be given the priority [33]. It shall maintain the speed following formulas (3)–(5) or decelerate based on the taxiway configuration.

If $S_{i,j}(t) \geq 0$, $f_j$ is further to $O_{i,j}$. As demonstrated in Figure 1, if $f_j$ should establish the visual separation by observing $f_i$, $f_i$ is the proceeding aircraft and $f_j$ is the following aircraft. The minimum lateral separation of taxiing aircraft is set to be $\Delta w$, and the lateral proximity rate at $t$ is defined as:

$$r_{i,j}^{\mathrm{W}}(t) = \frac{\Delta w}{D_{i,j}(t) + 1} \tag{7}$$

The minimum longitudinal separation is set to be $\Delta l$, and the longitudinal proximity rate of the two aircraft is defined as:

$$r_{i,j}^{\mathrm{L}}(t) = \frac{\Delta l}{S_{i,j}(t) + 1} \tag{8}$$

The proximity rate of the two aircraft is calculated as:

$$r_{i,j}(t) = \sqrt{(r_{i,j}^{\mathrm{L}}(t))^2 + (r_{i,j}^{\mathrm{W}}(t))^2}, \; r_{i,j}(t) \in (0, +\infty) \tag{9}$$

The above formula can quantify the proximity of the two aircraft.

If $r_{i,j}(t) \geq 1$, the proximity of the two aircraft is less than the longitudinal or lateral separation; therefore, the following aircraft $f_j$ shall regulate its speed as per $r_{i,j}(t)$.

### 3.3. Following Aircraft Speed Adjustment

The rate of deceleration of the following aircraft is defined as:

$$r_j^V(t) = (r_{i,j}(t))^{-0.1} \tag{10}$$

So $r_j^V(t) \in (0, 1)$ and there is a negative correlation with $r_{i,j}(t)$, the two aircraft would have a closer distance, and the following aircraft would have a greater deceleration rate:

$$v_j(t+1) = v_i(t) \times r_j^V(t) \tag{11}$$

If $r_{i,j}(t) \in (0, 1)$, the level of proximity of the two aircraft can meet the longitudinal and lateral separation; if $r_{i,j}(t) \in (0, 0.5)$, the two aircraft have a larger separation and the following aircraft would accelerate, as expressed in formula (12):

$$v_j(t+1) = \min(\min(v_C, v_T), v_j(t) + a_v) \tag{12}$$

After the speed adjustment, the aircraft would taxi forward:

$$L_i(t+1) = L_i(t) - v_i(t+1) \tag{13}$$

Thus, the proposed model simulates pilots' decision-making process by observing whether there is a merging intersection and conflict aircraft in front. Besides, the proximity rate and deceleration rate are dynamically derived according to the real-time position, speed, longitudinal and lateral separation. Regular patterns of acceleration and deceleration are revealed, making it possible to improve both safety and efficiency. The model can reproduce the establishing procedure of pilot-led visual separation, describe the separation establishment and conflict avoidance mode realized by the speed variation of the following aircraft and reflect the taxiing trends and mutual influence of aircraft.

## 4. Model Verification

### 4.1. Simulation Platform Set-Up

In this paper, the taxiing speed variation of aircraft at each step is dynamically calculated according to the taxiway configuration within the visual range and the position of the proceeding aircraft from the perspective of pilots. In this way, the taxiing process of aircraft under visual separation within period *T* can be simulated.

This model is achieved based on VC++. A junction of two taxiways is illustrated in Figure 1. The aircraft on both taxiways are heading to the intersection based on a certain proportion and taxiing separation. Taxiing speed and position of aircraft are validated by proposed model rules. The flowchart of simulation program is shown in Figure 2. The initial variable is set as $t = 0$, the total simulation time of *T* and the distance between the aircraft and the ends of taxiways of is $L_j(0)$. Visual separations between aircraft are investigated every second in the simulation following formulas (1) and (2). The routines of different simulation logics are executed accordingly, involving autonomous acceleration, deceleration before a turning point and speed adjustment by applying visual separation. The speed adjustment by visual separation can be divided into three steps: the proximity rate is calculated as per the longitudinal and lateral separations of the proceeding and the following aircraft; the expected deceleration rate is calculated; the operation of deceleration or normal acceleration is performed based on the deceleration rate. Finally, the position and distance of the two aircraft are derived from the speed adjustment.

### 4.2. Operation Process of Simulation Program

The ATC regulations of CAAC suggest that the minimum longitudinal taxiing separation of aircraft shall be 50 m and the wing separation during taxiing shall be no less than 10 m. If the aircraft is a common model from B737 or A320 series, its wingspan is about 35 m, the upstream taxiway length is set to typically 900 m and the included angle is set to 30° [34]. It can be obtained from the previous research that $v_C = 10$ m/s, $\Delta l = 50$ m, and $\Delta w = 80$ m [33], set $v^T = 4$ m/s, $a_v = 0.2$ m/s² [13,16,22]; the initial taxiing speed shall be 5 m/s if the two aircraft appear at the entrance of the taxiway concurrently.

Different scenarios in the simulation are presented in Figure 3. In this Figure, the proceeding aircraft 0008 is approaching the intersection and it decelerates to 4 m/s, while the following aircraft 0002 decelerates to 3 m/s to maintain the separation. The aircraft 0002 would decelerate accordingly when aircraft 0008 turns and decelerates. After deceleration, the longitudinal separation is 105 m and the lateral one is 122 m, so as to guarantee the safety of the two aircraft.

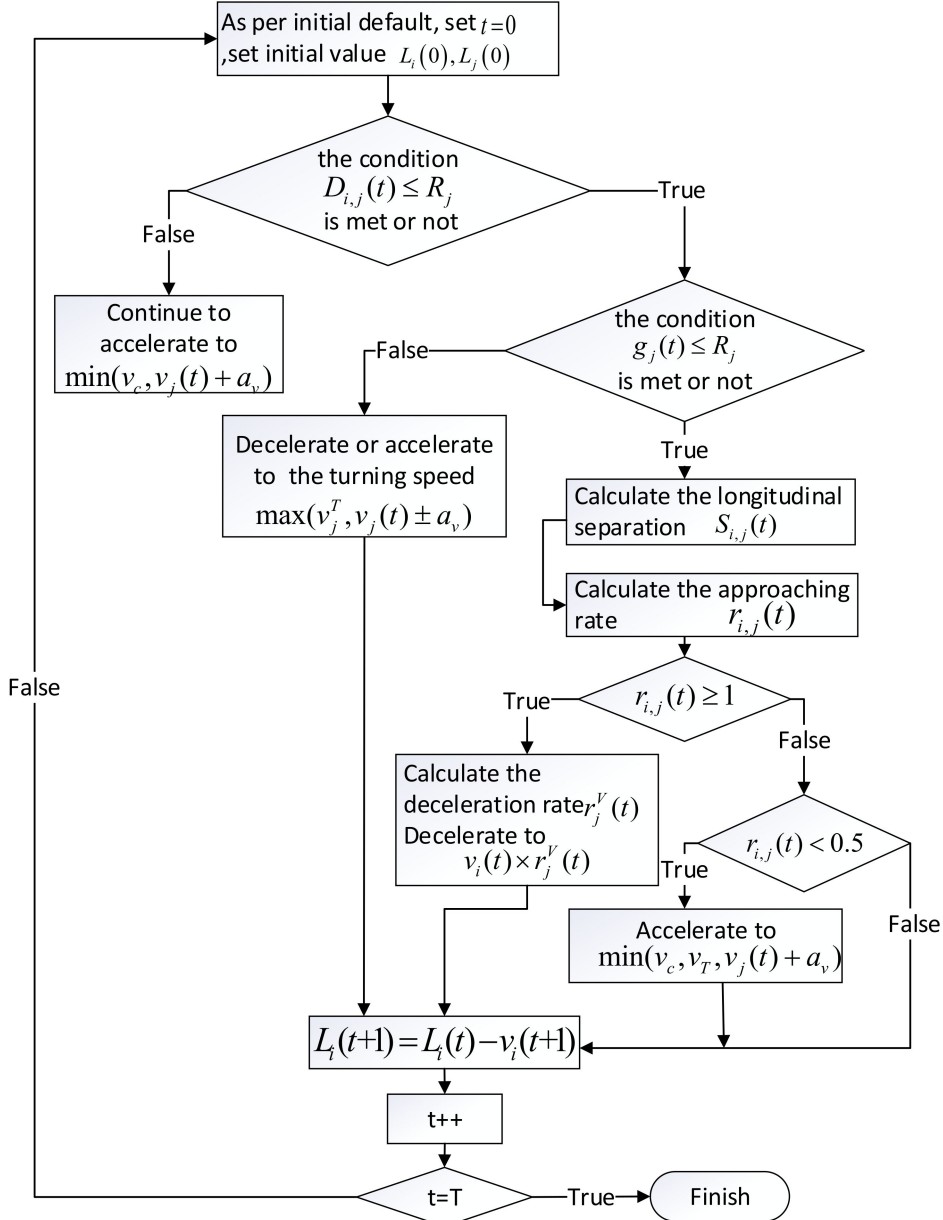

**Figure 2.** Flow Diagram of Model Simulation Program.

In Figure 3b, the proceeding aircraft 0017 accelerates to 10 m/s, with a longitudinal separation of 142 m and a lateral one of 160 m, while the following aircraft 0018 accelerates to 6 m/s at 0.2 m/s². This figure implies that the proceeding aircraft is fast, and the following aircraft is relatively slow. The proximity rate decreases with an increase in separation. Additionally, this figure demonstrates a corresponding acceleration of the following aircraft.

In Figure 3c, the proceeding aircraft 0024 decelerates to a turning speed of 4 m/s, with a lateral separation of 199 m and a longitudinal one of 189 m. Under the low proximity rate of the two aircraft, the following aircraft 0021 can accelerate to 8 m/s, reflecting the delayed deceleration process caused by the large separation between the two aircraft

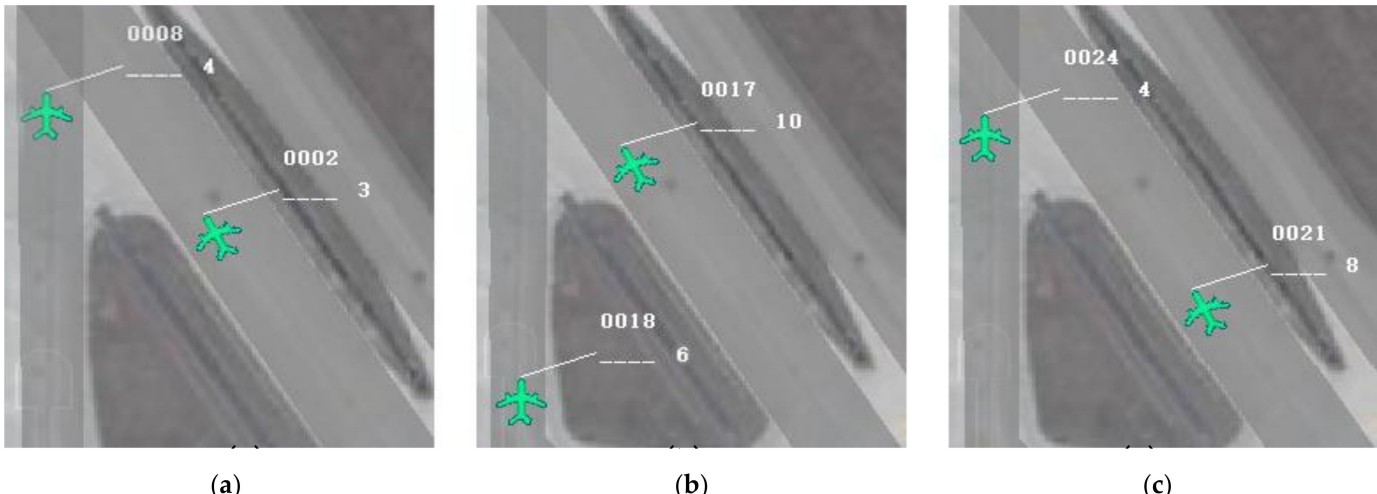

(a)  (b)  (c)

**Figure 3.** Schematic Diagram of Simulation Operation. (**a**) Following deceleration of the following aircraft; (**b**) Corresponding acceleration of the following aircraft; (**c**) the delayed deceleration process caused by the large separation between the two aircraft.

In this paper, the model has simulated the establishing process of visual separation under different moving trends and described the mutual influence of the aircraft moving process, as well as pilots' intention and operation to ensure safety and efficiency.

*4.3. Experimental Conclusions*

The key taxiing data such as taxiing speed, longitudinal separation, acceleration and proximity rate should be simulated and calculated to analyze the influence of visual range. $R_i$ and $R_j$ indicate the radius of the pilots' visual range; the larger the value, the wider the visual range. Set $t \in [0, 100]$ (hereinafter unit of $t$ is second, and metre for $R_i$ and $R_j$) with an increase of 1 s each time and $R_i = R_j \in [600, 900]$ with an increase of 3 m each time. Then, a total of $101 \times 101$ simulation operation values under the influence of different parameters were obtained.

The speed variation of the following aircraft under the visual separation is illustrated in Figure 4. If the $R_j$ is larger, visual range is wider, the following aircraft would realize the converging trend earlier, the deceleration timing would be earlier, and the deceleration would be less. The speed curve of $R_j \in [780, 900]$ exhibits a slow downward trend in this figure. However, $R_j \in [600, 700]$ suggests that the visual range is relatively narrow. Since the intersection and the position of the proceeding aircraft cannot be observed, $f_j$ would accelerate to 10 m/s. The following aircraft would decelerate sharply when the proceeding aircraft and the intersection are in sight. Accordingly, the curve would drop suddenly, with the maximum deceleration of 0.753 m/s$^2$, which is 3.75 times the comfortable deceleration [8–10]. As the separation between the two aircraft continues to be enlarged, the following aircraft gradually accelerates to reduce the separation, according to the rules for visual separation establishment defined in this paper. Due to the timely deceleration in the early stage, the following aircraft has sufficient acceleration time after 55 s when $R_j$ is large; the deceleration time is long with the acceleration period of less than 15 s when $R_j$ is small. Thus, the average speed is less than 5 m/s with low taxiing efficiency.

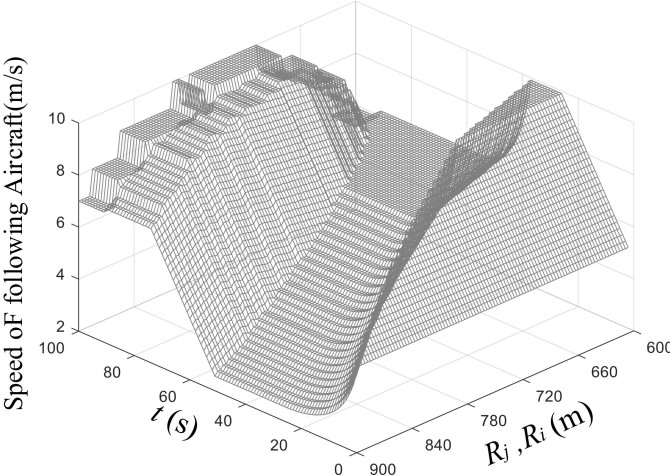

**Figure 4.** Trend Chart of Following aircraft's Speed with Time and Visual Range.

For that reason, the wider the visual range, the timelier the deceleration of the following aircraft, the slower the deceleration and the smaller and more stable the fluctuation of the taxiing speed. As a result, the operation efficiency can be improved.

The trends in longitudinal and lateral aircraft distance under the visual separation are presented in Figures 5 and 6. The longitudinal separation increases and then decreases with time. The smaller the value of $R_j$, the slower the increase rate of the longitudinal separation and the faster the decrease rate of the longitudinal separation. When $R_j = 600$, the longitudinal separation is 0 in the first 37 s and gradually increases to 173.8 m at 98 s. When $R_j = 900$, the longitudinal separation gradually increases to 125.2 m during the period of $t \in [1, 65]$ and then slowly decreases to 58.54 m, which is slightly larger than the minimum separation of 50 m and only 33.7% of the peak value in the time period of $R_j = 600$.

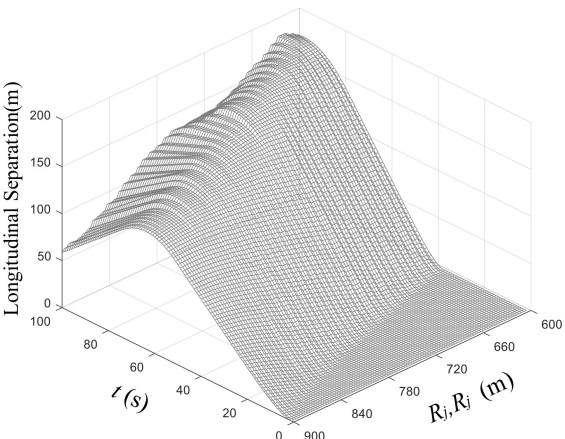

**Figure 5.** Trend Chart of Longitudinal Separation between Two Aircraft with Time and Visual Range.

As shown in Figure 6, the lateral separation gradually decreases. The pilot could discover the convergence after 30 s when $R_j$ is relatively small. Due to the continuous acceleration in the early period, the two aircraft are close to the intersection and the lateral separation decreases sharply. Besides, the decreasing trend of lateral separation is moderate as the following aircraft continues to decelerate. The range of lateral separation after 80 s is [173, 186] and the standard deviation is only 16.7 m, which is 22.8% of that of the first 50 s. However, the reduction of lateral separation slightly fluctuates when $R_j$ and $R_i$ are relatively large. When $R_i = R_j = 900$ and $t = 100$ s, the peak value of lateral separation is 225 m, since the proceeding aircraft could find the intersection earlier with a wider visual

range and decelerate to the turning speed according to formulas (4) and (5). Therefore, the two aircraft are far away from the intersection and have a large lateral separation.

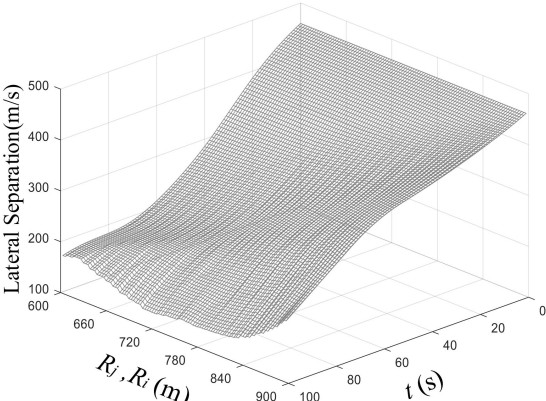

**Figure 6.** Trend Chart of Lateral Separation between Two Aircraft with Time and Visual Range.

The above 2 figures demonstrate that the following aircraft could regulate the speed and separation with a long time and distance when the visual range is wide, meeting the safety requirements and increasing the operation efficiency. The proceeding aircraft can observe the intersection in advance and reduce the speed, while the following aircraft could decelerate accordingly. The establishment of separation can be completed at a distance from the intersection to efficiently control the lateral conflict at the converging intersection with small angles.

The rules for visual separation establishment defined in this paper suggest that the proximity rate is the core parameter to judge the speed increase or decrease in the next second. The changing trend of the proximity rate of the two aircraft in 50–100 s is illustrated in Figure 7. The proximity rate is in inverse proportion to $R_j$ and $R_i$ when $t$ = 50 s; the proximity rate is in direct proportion to $R_j$ and $R_i$ when $t$ = 100. The following aircraft has established the separation in 1–49 s with a slow speed when $R_j$ and $R_i$ are relatively large; therefore, the proximity rate is less than 1. The proximity rate is less than 0.5, when $R_j$ and $R_i$ are larger than 840 m. Meanwhile, the acceleration of the following aircraft gradually increases the proximity rate. However, the following aircraft decelerates later with a higher speed when $R_j$ and $R_i$ are small; therefore, the proximity rate is still greater than 1 after 50 s, revealing that the separation between the two aircraft is less than the safety standard. Consequently, the following aircraft continues to decelerate to reduce the proximity rate. Additionally, the proximity rate is 0.941 after 100 s when $R_i$ = $R_j$= 900. According to the model rules, the following aircraft would still maintain its current speed after the next second. In Figure 5, the longitudinal separation is 58 m at this time.

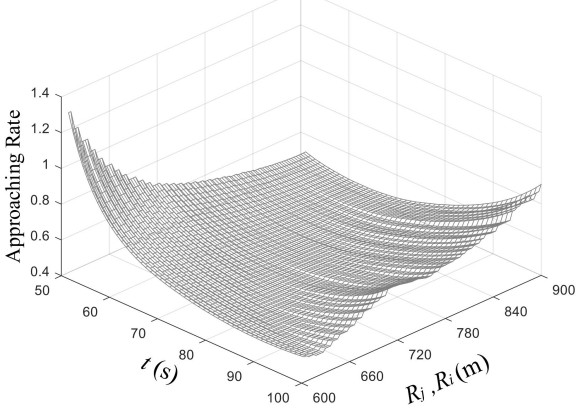

**Figure 7.** Trend Chart of the Approaching Rate of Two Aircraft with Time and Visual Range.

Therefore, the larger the values of $R_j$ and $R_i$, the smaller the fluctuation of the proximity rate in the later period, with the closing tendency of 1.0 at the end. This implies that pilots could make a timely observation, establish a smaller and safer separation between the two aircraft and keep taxiing at a constant speed. This process conforms to the actual operation and intention of pilots [8].

The taxiing conflict can be quantified with the method proposed in [8]. Two non-conflict situations:

$f_i$ leaves before $f_j$ enters the taxiway intersection, and the probability of conflict is:

$$P_j^i(t) = P\left(\frac{L_i(t) + l_i}{v_i(t)} < \frac{g_j(t)}{v_j(t)}\right) \tag{14}$$

Or $f_i$ enters the taxiway intersection zone after $f_j$ leaves, and the probability of conflict is:

$$P_i^j(t) = P\left(\frac{L_j(t) + l_j}{v_j(t)} < \frac{g_i(t)}{v_i(t)}\right) \tag{15}$$

The above two situations are mutually exclusive. So we have $P_{i,j}(t)$ as the conflict probability of aircraft $f_i$ and $f_j$ at time $t$:

$$P_{i,i}(t) = 1 - \left(P_j^i(t) + P_i^j(t)\right) \tag{16}$$

Figure 8 reflects that the conflict probability of crossing convergence between the two aircraft presents a trend of alternating increase and decrease with time. When $R_i = R_j < 760$ and $t$ is small, the conflict probability increases linearly and then shows an alternating increase and decrease trend, with a large increase and decrease range. There are three high conflict probabilities than 0.6 and two low conflict probabilities of less than 0.4. When $R_i = R_j > 820$, the overall fluctuation times and amplitude are small, with the maximum value of 0.54, regardless of increases or decreases in conflict probability. The conflict probability value is 0 when $R_i = R_j = 900$ and $t \in [55, 65]$. Further analysis reveals that the average value and standard deviation of the taxiing conflict probability decrease with the increase in the visual range. When $R_i = R_j = 600$, the average value of conflict probability is 0.566 and the standard deviation is 0.126, which is 2.79 times and 1.415 times, respectively, of that when $R_i = R_j = 900$.

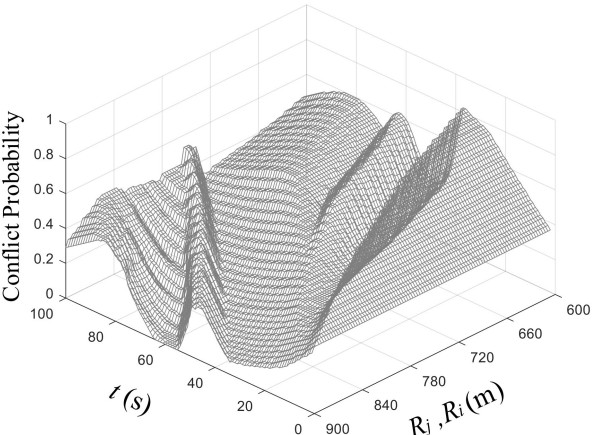

**Figure 8.** Trend Chart of Conflict Probability with Time and Visual Range.

## 5. Discussion and Case Studies

### 5.1. Simulation Results

According to the simulation data in the previous part, the following conclusions are drawn as follows:

1.  Conflict probability can be reduced to less than 0.5 after 50 s with visual separation. The conflicts can be reduced and the safety of taxiing can be improved by observing the proceeding aircraft and establishing visual separation at intersections during taxiing proposed in this research.
2.  Narrower visual range or lags in the operation of pilots can induce higher taxiing speed, with smaller horizontal separation and more conflict. Although the pilot of the following aircraft takes a long and rapid deceleration to lower the conflict probability in the later period, it would cause the fluctuation in conflict probability, poor taxiing stability and low operation efficiency.
3.  The relatively wide visual range and the timely operation of pilots can quickly control the separation reduction and converging taxiing situation. The speed adjustment of following aircraft is slight and infrequent and the conflict probability and taxiing process are relatively stable with relatively high operation efficiency.

### 5.2. Model Validation Using Empirical Data

The above analysis was calibrated and validated using empirical data of taxiing trajectory from a hub airport in Southern Central China. The converging taxiing trajectory of aircraft at key intersections was investigated, which was then compared with the taxiing model proposed in this paper. Figures 9 and 10 present five clusters of aircraft converging trajectory at the same intersection, which show that the speed variations of the proceeding and following aircraft within the first 75 s before passing through the intersection.

As shown in Figure 9, the pilots of the following aircraft observe other aircraft on the left, predict the conflict and decelerate with an average deceleration of 0.35 m/s$^2$ in the first 30 s. In Figure 10, the proceeding aircraft gradually accelerate with an average acceleration of 0.1 m/s$^2$ in the first 30 s. This implies that pilots of the two aircraft can independently adjust the taxiing speed by observing the positions of the front intersection and the conflict aircraft before meeting, with the following aircraft decelerating and the proceeding aircraft accelerating. In this way, the two aircraft can achieve longitudinal and lateral safety separation before meeting. Therefore, the taxiing process described in this paper based on the real-time operation of pilots and the establishment of visual separation is consistent with the actual taxiing trajectory.

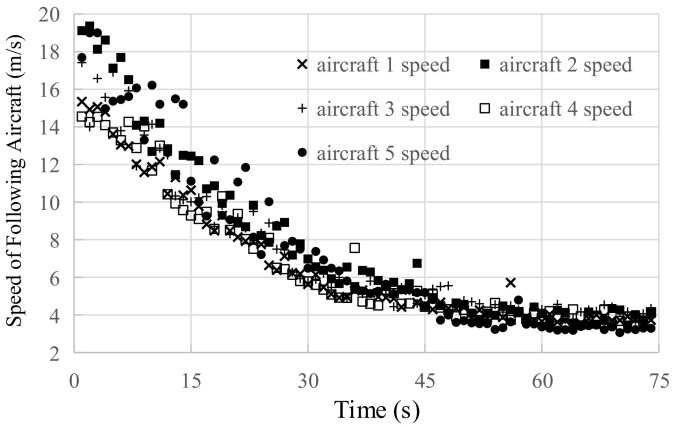

**Figure 9.** The Taxiing Speed of the Following Aircraft.

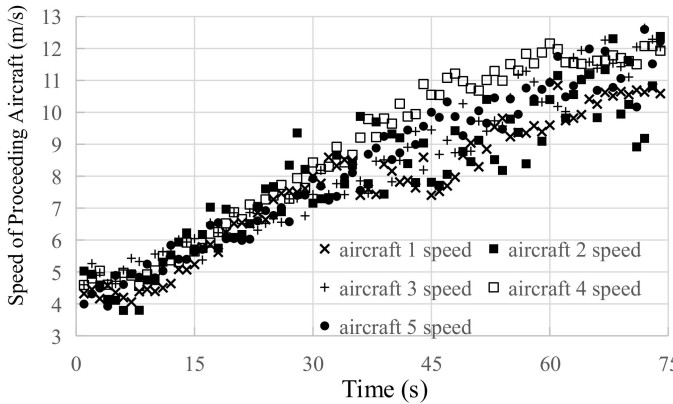

**Figure 10.** The Taxiing Speed of the Proceeding Aircraft, Aircrafts are not the same ones in Figure 9.

Figure 11 demonstrates the average value of the above five groups of the aircraft's speed and the taxiing speed simulated by the model designed in this paper. This simulation set $R_l = R_j = 800$, the initial speed of aircraft 1 of 16.2 m/s, the initial taxiing speed of aircraft 2 of 4.2 m/s, $\Delta l = 65$ m, $\Delta w = 80$ m [34], $v^{\mathrm{T}} = 12$ m/s, and $a_v = 0.2$ m/s$^2$ [13]. Figure 12 exhibits the simulated taxiing trajectory points of aircraft.

Figure 11 suggests that the changes in simulated taxiing speed are the same as the overall trend of actual trajectory average speed. According to the simulation data, the deceleration of the following aircraft is high in the early period and low in the later period and it passes through the intersection at a constant speed after 60 s. The proceeding aircraft gradually accelerates to the average speed of 0.2 m/s$^2$ in the first 30 s and maintains taxiing at a constant speed. When the front intersection is observed, it continues to accelerate to the maximum speed of 12 m/s at the intersection. The comparison reveals that the acceleration is low when pilots accelerate continuously in actual operation, while the model in this paper adopts a higher acceleration to form a phased acceleration. In Figure 12, the trajectory points of the following aircraft are changed from sparseness to denseness and the trajectory points of the proceeding aircraft are changed from denseness to sparseness. This reflects that the two aircraft have been adjusted at separation after deceleration and acceleration and the pilots of the following aircraft have established the visual separation for both aircraft. In this paper, the simulation results of this model are consistent with the actual operation process.

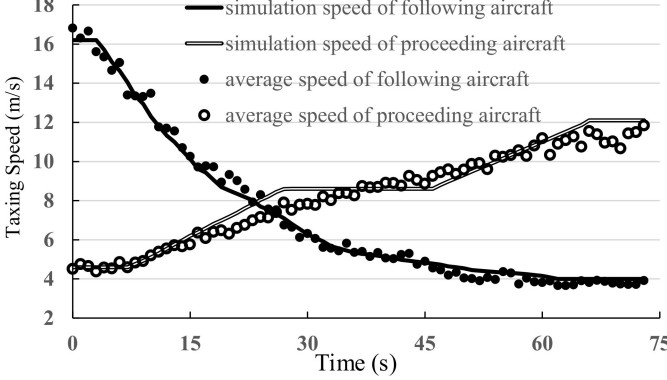

**Figure 11.** Average Speed of Actual Trajectory and Simulated Trajectory Speed, Proceeding aircraft should be gradually accelerated to the comfortable taxiing speed of $v_{\mathrm{C}}$, and then gradually accelerated to the turning speed limit of $v^{\mathrm{T}}$, so comes the plateau in line of simulation speed of proceeding ac.

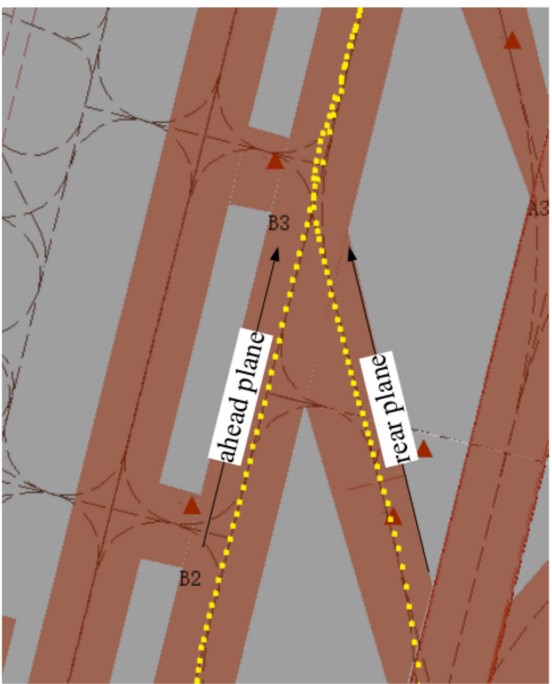

**Figure 12.** Simulated Taxiing Trajectory.

The following table list the history data of two aircraft departure from the hub airport, which includes callsign, date time of departure, taxi time and number of plots of each aircraft. Number of plots is number of position data records of each aircraft obtained from airport surface surveillance radar. A taxiing conflict was discovered in the data.

Figure 13 illustrates the trajectories of the aircraft MAS377 and CSN3501 (Table 1) converging and taxiing towards the 90° intersection when the visual range is limited by the occlusion of the parked aircraft (hereinafter referred to as case I). Figure 14 presents the simulated taxiing trajectories of the two aircraft at intersection when the visual range is not limited (hereinafter referred to as case II).

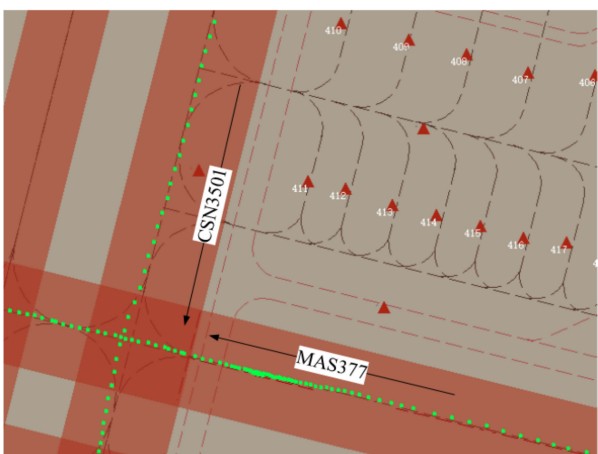

**Figure 13.** Actual Taxiing Trajectory in Case I.

**Table 1.** Trajectory data information.

| CallSign | Dep Date Time | Taxi Time | Number of Plots |
|----------|---------------|-----------|-----------------|
| MAS377 | 14 January 2014 16:06:24 | 2257 s | 1481 |
| CSN3501 | 14 January 2014 16:15:16 | 1270 s | 718 |

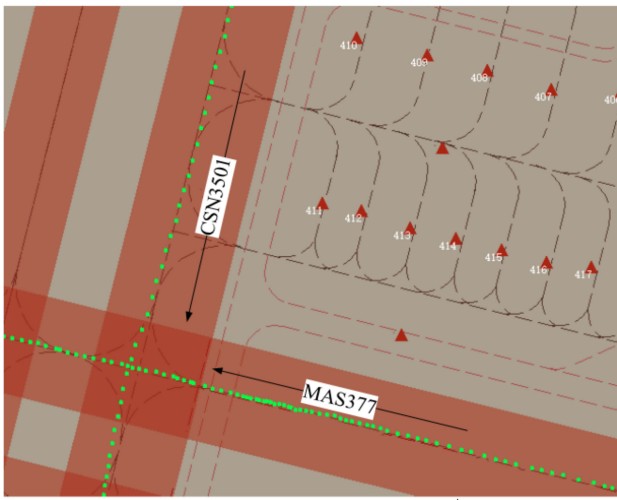

**Figure 14.** Simulated Taxiing Trajectory in Case II.

The comparison between Figures 13 and 14 reveals that blind areas for taxiways are generated for the presence of visual occlusion since some aircraft are parked at aircraft stands 411–417. The pilots of the two aircraft cannot conduct visual observation when the distance is far away. Therefore, MAS377 taxis at a larger speed for a longer time and the deceleration timing is later. It is less than 90 m away from the intersection when the pilots of MAS377 see the front CSN3501. To prevent collision, the pilots decelerate sharply and make the aircraft at a rest to hold for CSN3501 to completely pass through the intersection before continuing taxiing. The trajectory of MAS377 exhibits significant point aggregation (Figure 13). Figure 14 implies that the visual range is not limited and the visible range is 800 m. Pilots of MAS377 observe CSN3501 at 620 m from the intersection and gradually decelerate, with the deceleration to 1.5 m/s at 150 m from the intersection and then gradually accelerate after the proceeding aircraft passes through the intersection. The comparison between the two figures demonstrates that the pilots of the following aircraft can see the proceeding aircraft in time when the visual range is wide and then establish the taxiing separation by decelerating in advance.

The speed variations of MAS377 under different conditions are presented in Figure 15. As revealed by the comparison, the visual range in Case I is narrow. The initial deceleration of MAS377 is only 0.16 m/s$^2$ and the proceeding aircraft decelerates sharply after 26 s. The average deceleration within 7 s is 0.67 m/s$^2$, with the maximum deceleration of 1.16 m/s$^2$, which is 5.8 times the comfortable deceleration. The holding time of MAS377 is about 20 s to prevent collision. The visual range in Case II is wide. The pilots of MAS377 establish the visual separation according to the position of CSN3501 and decelerate step by step. After 20 s, MAS377 could decelerate to 5 m/s with an average deceleration of 0.24 m/s$^2$. As CSN3501 approaches the intersection continuously, MAS377 decelerates to 1.5 m/s with an average deceleration of 0.13 m/s$^2$ after 26 s. Therefore, the pilots can see the proceeding aircraft and convert standstill into slow taxiing by timely deceleration under the condition that the visual range is wide and there is no blind area, so as to effectively reduce the rapid deceleration and interrupted holding during taxiing.

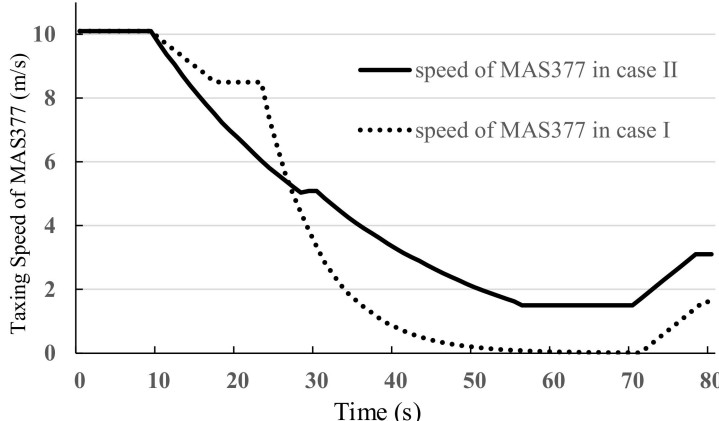

**Figure 15.** Taxing Speed Variations of MAS377 under Different Conditions.

The conflict probability of the two aircraft under different conditions is presented in Figure 16. In Case I, the conflict probability reaches a peak value of 91.4% at 28 s, indicating the relatively high collision risk of the two aircraft. As the MAS377 rapidly decelerates to 1.5 m/s, the conflict probability decreases to 52%, and the conflict probability is less than 5%when the MAS377 is holding. In Case II, the maximum conflict probability is only 67.7%. Compared with Figure 15, the conflict probability starts to decrease immediately when MAS377 starts to decelerate, suggesting that the following aircraft can mitigate the conflict risk by a slight deceleration when the two aircraft are far away from the intersection. The conflict probability is reduced to 38.5% at 28 s, implying that the earlier establishment of the visual separation can contribute to more effective control of the risk of operational conflict, thus the safety of taxiing can be improved.

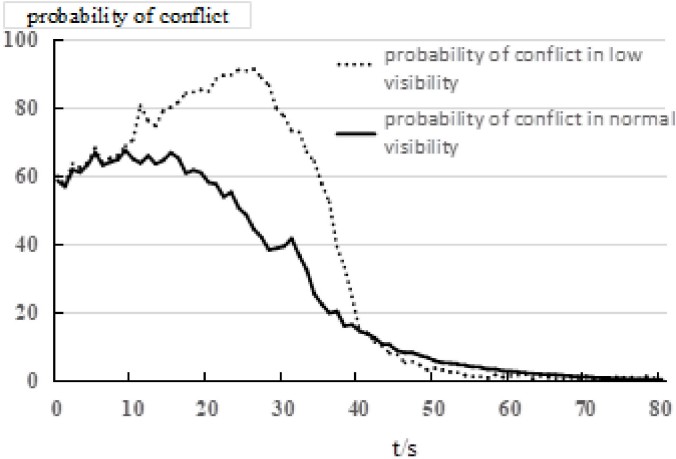

**Figure 16.** Conflict Probability under Different Conditions.

Hence, a light indication system and conflict hot spot signs shall be used in the actual operation of the airport to indicate the location of intersections [5]. Moreover, airborne equipment such as an electronic map package needs to be adopted to display the taxiway structure and monitoring information [26–28]. Simultaneously, the controller should inform the positions and moving trends of conflict aircraft at the key positions in time and remind pilots to take timely measures. The above equipment and means can help the pilots establish situational awareness, expand the scope of the pilots' visibility, eliminate the blind spots, increase the observation range to assist the pilots to establish the visual separation, so as to reduce taxiing conflicts and improve operation efficiency.

## 6. Conclusions

Designing a model to finely describe the aircraft taxiing process and identify the key factors influencing airport surface operation safety and restricting operation efficiency is a hot spot in aviation research. In this paper, the process of pilot observing taxiway configuration to initiate a visual separation is analyzed. The proximity rate between the taxiing aircraft and the deceleration rate of the following aircraft are defined and quantified. Finally, an aircraft taxiing simulation model based on visual separation was proposed. The simulation model is established according to the ATC control regulations of CAAC and actual operation data.

The data simulation on the computer-program-implemented model is conducted to quantitatively analyze the impact of visual range and the pilots' adjustment on taxiing separation and conflict. The conclusions can be drawn as follows.

1. This model can simulate the process of establishing visual separation, during which the pilots adjust aircraft taxiing speed and predict possible conflicts timely with the position of proceeding aircraft and the intersection as references.
2. This model can reproduce the taxiing process during which the following aircraft decelerates and accelerates alternately to improve the operation efficiency and enhance the safety of taxiing.
3. The pilots can effectively reduce the risk of conflicts and improve the safety of taxiing by observing the proceeding aircraft and intersections and then initiating the visual separation during taxiing. The pilots shall be informed of the position of proceeding aircraft by the indication of lighting system, controllers' reminder and other means when the visual range is less than 100 m or there is occlusion. Enlarging the value of visual range on purpose can effectively reduce the conflict risk and improve taxiing stability.
4. The simulation results of this model are consistent properly with the actual operation program and real trajectory. The conclusions in this paper are capable of evaluating the impact of pilots' operation and visual range, and can provide technical support for airport risk evaluation and navigation strategy on airport surface.
5. This model can be applied to simulate conflict detection at X, T, Y and + shaped intersection.

**Author Contributions:** Conceptualization, K.Y. and R.K.; methodology, K.Y.; Project administration, J.Z.; software, K.Y.; validation, R.K.; investigation, K.Y.; resources, H.Y.; data curation, R.K.; writing—original draft preparation, K.Y.; writing—review and editing, R.K.; supervision, H.Y. and J.Z.; funding acquisition, R.K. All authors have read and agreed to the published version of the manuscript.

**Funding:** This research was funded by [Key Research and Development Projects of Sichuan Province] grant number [2021YFG0171] and [2022YFG0196]. And Key Projects of Civil Aviation Flight University of China (ZJ2021-05).

**Institutional Review Board Statement:** Not applicable.

**Informed Consent Statement:** Not applicable.

**Data Availability Statement:** Not applicable.

**Conflicts of Interest:** The authors declare that they have no conflict of interest.

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
