# Peer review of "Effects on Taxiing Conflicts at Intersections by Pilots’ Sensitive Speed Adjustment"

_aerospace, doi:10.3390/aerospace9060288_

Round 1
Reviewer 1 Report
This paper proposes a theoretically-based simulation model based on procedural stipulations and pilot visual range capabilities to assess surface aircraft convergence, changes to speed, and taxiing priority. It is interesting contribution, but there are opportunities for improvement.
Overarching
While this work includes pilot’s visual range performance, I’m not sure if there is an implicit assumption of how fast aircraft are moving. I also wonder what the implications/assumptions are regarding how often pilots make the judgement of where the other aircraft is, and whether they are making them at the same time -so their judgements are coordinated.
Terms
- Pilots’ Visual Distance = Pilots’ distance perception?
- Pilots’ Observation Distance = same as above?
- pilot's visual separation procedure
- “pilot taxi observation”.. behavior
Keyword: Pilot, human performance model
Introductions
There is a question in the review about whether there is inappropriate self-referencing, and I do not find that. However, there is not adequate literature review of others’ foundational work in this area to be found in open literature. I suggest searching for the work of Gary Lohr, Lynda Kramer, Lawrence Prinzel at NASA Langley, for example.
Line 58 – insert “taxiing” in front of trajectory. Do you mean speed and path? Trajectory here I think is the wrong word – it implies a projectile flying.
Line 67 – “few research’ – little research, or few research papers
Look for opportunities to remove ‘therefore’
Line 76 - average accuracy was about 30s.; what is that in terms of percent error?
Line 100 – “since most airports are not equipped 100 with surface guidance and control systems [23].” This citation is 10 years old, so the observation is likely at least 12. Is this still true?
Line 115 – “which weakens the pilot's driving behavior” – I’m not sure what is meant by this? Reduces the predictability of ?
Line 118 - “pilot's visual separation procedure” – visual separation requirements? Or pilot’s visual separation judgement?
Figure 1 – I would suggest thinking about how to format this figure such that a photocopy of it is still understandable. It looks to me that the red and blue areas are not distinguishable except by color in the overlapping area.
Line 142 and throughout – “flight paths” should be “taxiing paths” – if they are on the ground.
Lines 138 etc – Pilots’ judgement of convergence depends not only on the ability to see other aircraft, but on their judgement of speed.
158 - tendency of convergence = rate of convergence?
169 – “If condition (1) is not met, the intersection cannot be observed” – by the red aircraft. Wouldn’t’ this be condition 2? Aren’t these switched? (lines 169-172)
170 etc – terms in these equations have not been defined. V(c), v^T, a(v),
184 – “longitudinal separation” - longitudinal separation (Si,j)
Is there an assumption in the prioritization that they are moving at the same speed?
190 As demonstrated in Figure 1. Rules for Visual Separation Establishment in Aircraft Taxiing., if j f should establish the visual separation by observing i f , if i f is the proceeding aircraft and j f is the following aircraft. – I am having trouble understanding this. Please put all conditions (if’s) in one comma-delineated list and then a “; then” before indicating the consequence.
Why is minimum lateral and longitudinal separations of taxiing aircraft – a delta?
Equation 10 – appears to indicate that the ri,j is raised to the minus .1; is that really correct? Should it be (0,1)
Please put Figure 2 in section 3.0 before 3.1. This would be so helpful. It would also help the reader to have Figures 1 & 2 on the same page if that is possible.
Section 4.3 please indicate units on your condition ranges.
You indicate the consequences of the extent of visual range perception, but there are consequences of the angular displacement of visual situation awareness as well – how far to the side can you see?
Line 403 – Aircraftpresent = Aircraft present
Line 405 = remove “first”
I am finding Figures 9&10 confusing. Are the aircraft numbers on both figures meaning the same aircraft? That is, is Aircraft 1 in Figure 9 the same aircraft as Aircraft1 in Figure 10 and these are showing the speeds when each aircraft is in each role? If that’s the case, would it be in Figure 9? Where it is not following anyone? Similarly would Aircraft 5 be in figure 10? Or, are they just 5 different aircraft in each figure?
The fit to the following aircraft looks to be of higher order than the fit to proceeding aircraft. Can you explain the inflection points in the latter?
Line 491 – reference not found – lost link – no title on the suspect Figure below that looks like it might be the reference.
The untitled figure for p(conflict) – this is saying that there is more of a p(conflict) when you can see better? I would expect the other way around, and that seems consistent with your statement: implying 498 that the earlier establishment of the visual separation can contribute to more effective 499 control of the risk of operational conflict, thus the safety of taxiing can be improved..
The Abstract promises “Additionally, the influences of pilot taxi observation and autonomous speed regulation behavior on risk level are discussed under the different visual conditions.” But I did not see a great deal of interpretation with regard to this and what the information requirements (what data, how timely, how precise) would be to support it.
Check reference formatting for citations – spaces missing or added: 13, 26, 29. And in general, typically source and volume are emphasized, but leave that to the editor.
Author Response
No.aerospace-1726731
5,21,2022
Dear Reviewer,
We would like to thank you for your efforts in reviewing our manuscript titled "Effects on Taxiing Conflicts at Intersections by Pilots’ Sensitive Speed Adjustment", and providing many helpful comments and suggestions, which will all prove invaluable in the revision and improvement of our paper, as well as in guiding our research in the future.
We have studied your comments point by point, revised the manuscript accordingly. The amendments are highlighted in "Track Changes" in the revised manuscript. All authors have approved the response letter and the revised version of the manuscript.
Please see the attachment.
We hope that the revised version of the manuscript is now acceptable for publication in Aerospace. If you have any queries, please do not hesitate to contact me.
Thank you again for your valuable comments and suggestions. I look forward to hearing from you soon in due course.
Yours sincerely,
Kai Yang

Reviewer 2 Report
The paper summarizes a simulation model based on pilot visual separation for the purposes of maintaining both visual separation and efficiency during taxi procedures at airports. The model is set up to simulate movement on the airport surface under different motions of the aircraft and pilot intentions. The authors' results indicate that taxiway conflicts can be reduced to 50% based on visual separation adjustment of 50 seconds when the convergence angle is 30 degrees. The article is fairly easy to follow.
The topic of increasing airport efficiency is relevant and I recommend publication after addressing some minor points. Specifically, I would like to see some commentary as to how the underlying assumptions about deceleration rate in this work would be effected by both pilot reaction time and weather (ie, a wet or slippery runway for example). I would also like the calculation for conflict probability to be more explicitly defined in the paper. In addition, there are a few more specific edits for the authors to consider:
A few times in the paper the figure number and caption are both written when referencing a figure within the article text (for example, line 138). These references should be the figure number only.
Line 14-16: this sentence is very long and is a bit difficult to follow. Consider dividing the sentence.
Line 71: existing research "has" widely
Recommend discussing Figure 1 in the text before the figure is shown in the text.
Line 117-120: this sentence is very long and is a bit difficult to follow. Consider dividing the sentence.
Line 264, 439, 477-478, 483-484: the seconds should have the number 2 in the superscript m/s^2
Line 284: a fragment, consider revising
Line 295-296: Please provide a reference for the deceleration rate referenced here
Line 316, 320, 331, 355, 371, 374, 376, 379-380: please add units to Rj and Ri
Figure 6: Please check your units and labeling in the figure. I assume that lateral separation should be in m. Also the time and Ri, Rj appear to have switched axes compared to figure 5.
Line 347: please add units to t
Line 453: Please introduce in the text before showing it in the paper. Also please define "number of plots" in table 1.
Table 1: I would like to see more discussion leading into the choice of more detailed study of these two aircraft in particular.
Line 456: Missing a value after "two aircraft at..."
Line 466: Table1Figure 14 seems like a typo. Please address
Line 491: Reference missing; please address
Figure 16 is missing a caption
Author Response

(The authors gave the same response as above.)

Reviewer 3 Report
Please see the attachment.

Author Response

(The authors gave the same response as above.)
